# Barriers and facilitators to Water, Sanitation and Hygiene (WaSH) practices in Southern Africa: A scoping review

Nkeka P. Tseole[1]*, Tafadzwa Mindu[1], Chester Kalinda[2,3], Moses J. Chimbari[4]

**1** School of Nursing and Public Health, College of Health Sciences, University of KwaZulu-Natal, Howard Campus, Durban, South Africa, **2** Bill and Joyce Cummings Institute of Global Health, University of Global Health Equity (UGHE), Kigali, Rwanda, **3** Institute of Global Health Equity Research (IGHER), University of Global Health Equity (UGHE), Kigali, Rwanda, **4** Department of Behavioural Science, Medical and Health Sciences, Great Zimbabwe University, Masvingo, Zimbabwe

\* nkekathabiso@gmail.com

**Data Availability Statement:** All relevant data are within the paper and its Supporting Information files.

## Abstract

A healthy and a dignified life experience requires adequate water, sanitation, and hygiene (WaSH) coverage. However, inadequate WaSH resources remain a significant public health challenge in many communities in Southern Africa. A systematic search of peer-reviewed journal articles from 2010 –May 2022 was undertaken on Medline, PubMed, EbscoHost and Google Scholar from 2010 to May 2022 was searched using combinations of predefined search terms with Boolean operators. Eighteen peer-reviewed articles from Southern Africa satisfied the inclusion criteria for this review. The general themes that emerged for both barriers and facilitators included geographical inequalities, climate change, investment in WaSH resources, low levels of knowledge on water borne-diseases and ineffective local community engagement. Key facilitators to improved WaSH practices included improved WaSH infrastructure, effective local community engagement, increased latrine ownership by individual households and the development of social capital. Water and sanitation are critical to ensuring a healthy lifestyle. However, many people and communities in Southern Africa still lack access to safe water and improved sanitation facilities. Rural areas are the most affected by barriers to improved WaSH facilities due to lack of WaSH infrastructure compared to urban settings. Our review has shown that, the current WaSH conditions in Southern Africa do not equate to the improved WaSH standards described in SDG 6 on ensuring access to water and sanitation for all. Key barriers to improved WaSH practices identified include rurality, climate change, low investments in WaSH infrastructure, inadequate knowledge on water-borne illnesses and lack of community engagement.

## Introduction

Inadequate water, access to improved sanitation, and hygiene (WaSH) are global health challenges affecting about one-third of the world's population [1, 2]. Improved sanitation and hygiene are essential because they reduce environmental health risks [3]. Global diarrheal

**Funding:** This research was funded by the National Institute for Health Research (NIHR) Global Health Research programme (16/136/33), UK and the University of KwaZulu-Natal. We also acknowledge University Administration Support Program (UASP) funding for this manuscript.

**Competing interests:** There are no competing interests for this manuscript.

disease statistics show that more than one million annual deaths are related to poor WaSH practices as over one-third of the world's population do not have basic sanitation [4]. Although adequate WaSH coverage is critical for improving quality of life, globally about 2 billion people do not have access to clean water [5] and over 263 million people walk long distances to collect water from rivers, streams and lakes. Furthermore, at least 159 million people drink water from unsafe sources [5].

In Africa, about 70 percent of rural water schemes are non-functional or intermittently functional at any given time [6] resulting in compromised human wellbeing [7]. Due to poor WaSH practices in Africa, about 28 percent of the population in the region still practice open defecation [1]. Unsafe sanitation behaviours are responsible for around 775, 000 world deaths annually of which 5 percent are in low-income countries [1]. Universal, affordable, and sustainable access to WaSH is one of the key public health and developmental issues. Plans to improve WaSH coverage are instituted in the Sustainable Development Goals (SDG) goal 6 which seeks to ensure availability and sustainable management of water and sanitation for all by 2030 [8]. Even though this SDG advocates for progressive reduction of inequalities related to hygiene and universal access to clean water and sanitation [8], continued inequalities in access to clean water and improved sanitation between rural and urban settings are still a challenge [8–11].

Improved WaSH practices have the potential to reduce the prevalence of diseases such as schistosomiasis, cholera, diarrhea, polio, and typhoid which are prevalent in most sub-Saharan African countries. However, people still lack adequate information on WaSH leading to poor sanitation and hygiene practices. Southern Africa is among regions with very low rates of WaSH coverage in the world [8]. The provision of clean water to most rural communities in Southern Africa is insufficient and this exacerbates challenges associated with sanitation and hygiene [12]. For instance, hand washing is a cost-effective and simple approach used for the control of water-based infections and yet despite its simplicity and effectiveness it is not widely used [13].

Mitigating inequalities linked with access to WaSH is therefore critical. Understanding patterns of inequalities in WaSH practices, and how these are influenced by different facilitators and barriers is vital to providing effective interventions to mitigate inequalities in WaSH coverage in Southern Africa. Using a scoping review guided by the methodological framework for scoping, we examined facilitators and barriers to effective WaSH practices in Southern Africa and identified knowledge gaps on the same [14].

## Materials and methods

### Design

We conducted a scoping review of published peer-reviewed articles on barriers and facilitators to WaSH practices in Southern Africa. The use of scoping review studies allows researchers to identify and analyze existing evidence from published peer-reviewed journal articles related to specific research areas. Scoping reviews are conducted to understand the status of knowledge related to a topic of interest [14]. Our review included studies published from 2010 to May 2022 and was guided by Arksey and O'Malley's 2005 scoping review framework which describes six stages: (1) identifying the research question; (2) identifying relevant studies; (3) selecting studies; (4) recording data; (5) organizing, summarising and reporting the results and (6) consultation exercise [14]. The optional six step is usually conducted with key stakeholders to inform and validate study results [14]. We did not include that in our review.

## Search strategy

Our review focused on peer-reviewed journal articles, both quantitative and qualitative studies published from 2010 to May 2022 to identify facilitators and barriers to WaSH practices. We conducted a systematic electronic search of peer-reviewed journal articles from various databases including PubMed, EbscoHost, Medline and Google scholar using the following keywords: "*facilitators; barriers*; *water*; *sanitation*; *hygiene*; *WaSH practices* and *Southern Africa*." Using the keywords, we developed "index terms" by combining keywords and their synonyms and used the Boolean operators "AND", "OR" and truncations to create search strings: "*Water AND sanitation AND hygiene AND Facilitators (AND motivators) AND barriers (OR hindrances) AND WASH practices AND Southern Africa*". After eliminating all the duplicates for extracted articles, we identified relevant articles by screening the titles and abstracts. Full articles of the selected titles and abstracts were selected for eligibility. These articles were further screened (full-text) for relevance in terms of their focus and aims.

## Inclusion and exclusion criteria

The review included articles describing interventions on WaSH practices in Southern Africa with a particular focus on facilitators and barriers. Articles included in the study were published in the English language from 2010 to May 2022. We excluded reviews, i.e. systematic, scoping and meta-analysis that were published before 2010. Our review also excluded reports, working papers and articles published before 2010. Our exclusion criteria further excluded articles that were published in other languages other than English.

## Quality assessment

We assessed all selected articles for quality using a mixed methods appraisal tool (MMAT) [15]. MMAT is used as a tool to appraise the quality of different study designs [15]. For each study, we used scores ranging from 0 to 10, where 0–4 = "Low" quality, 5–7 = "Moderate" quality and 8–10 = "High" quality. The majority of the articles selected scored moderate. No studies scored "Low", 17 articles scored "Moderately" and one article scored "High". Indicators used for quality scores included: (a) a clear definition of the study objective and aim, (b) study design appropriate for stated aims, (c) justified sample size, (d) targeted population defined, (e) risk factor and outcome variables measured, (f) methods clearly described, (g) study results described, (h) discussions and conclusions justified, (i) study limitations discussed and (j) ethical approval for the study attained.

## Data extraction and analysis

In the data extraction phase, a total of 18 articles were selected (Fig 1) based on the inclusion and exclusion criteria. All records were downloaded using Zotero software and duplicates were removed. We created a data extraction table (Table 1) that captured the following information: authors, year of publication, objectives of the study, the type of the study, geographical location from where the studies were conducted and the summary of the main findings from each study.

# Results

Our electronic search from PubMed provided 1252 records, EbscoHost 62 records and 75 records from Google scholar. The electronic title search provided a total of 1389 articles (Fig 1) from which 24 duplicates were removed. One thousand, three hundred and one (1301) articles were deemed illegible and were removed after screening their titles. Sixty-four (64) articles

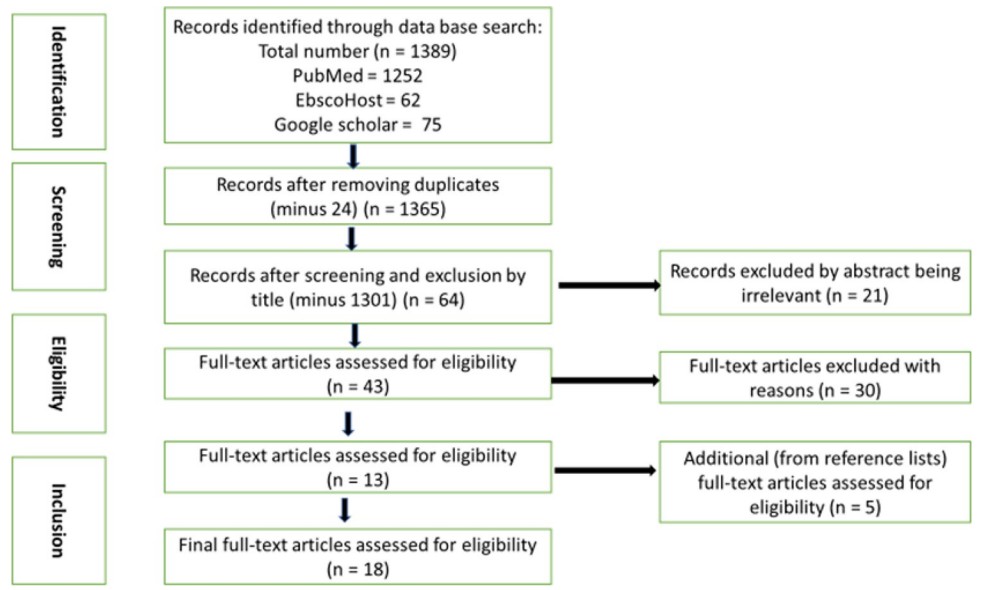

**Fig 1. PRISMA flow diagram showing steps followed to select articles.**

that remained were screened based on their relevance by abstracts and of these, twenty-one (21) articles were removed. Full-text screening for the remaining 43 articles was done and 30 articles were removed due to irrelevant focus and aim concerning the objective of this review. Among those removed, one article covered a scope outside Southern Africa, another article used secondary data collected between 1995–2006 although the paper was published in 2015. One article was a working paper, and the other excluded studies were reports, systematic and scoping reviews. We remained with 13 legible records deemed relevant. Five (5) additional records were identified from the reference lists of eligible articles and were included as grey literature for full-text review resulting in a total of 18 articles (Fig 1).

## Characteristics of the selected articles

**Distribution by country.** Out of 18 articles reviewed, most (n = 5, 27%) of the studies were conducted in Zambia while from Botswana, Lesotho, Mozambique, South Africa and Zimbabwe, ten studies (two studies from each country) were reviewed (Table 1). Three studies (one from each country) were from Malawi, Eswatini and Namibia. Six studies were quantitative [16–20], four were qualitative [21–24], while nine used mixed methods approach [25–27, 29–32].

**Barriers to WaSH practices.** The key themes that emerged with regards to barriers to WaSH practices in Southern Africa from the articles reviewed comprised (a) inadequate financing, (b) population growth, (c) inadequate knowledge of waterborne diseases, (d) ineffective local community engagement in WaSH interventions, and (d) climate change.

**Inadequate financing.** Lack of skilled personnel and poor laboratory equipment was reported to compromise the quality of water and water supply services owing to insufficient funds [19]. The situation compromises clean water supply, and resulting in poor sanitation and hygiene practices [19, 22]. Due to insufficient funding, in some places where there was WaSH infrastructure in place, there was poor or no maintenance on the damaged infrastructure. The challenge of broken WaSH infrastructure contributes negatively to improved sanitation and hygiene practices. Inadequate funding led to inadequate WaSH infrastructure

**Table 1. Summary of studies used in the review.**

| Author(s)/year of publication | Title of the study | Objective(s) of the study | Type of the study | Country | Facilitators for WASH | Barriers for WASH |
|---|---|---|---|---|---|---|
| Tubatsi, G., Bonyongo, M.C. & Gondwe, M. (2015). | Water use practices, water quality, and households' diarrheal encounters in communities along the Boro-Thamalakane-Boteti river system, Northern Botswana | Assessing river water quality and water use patterns in selected communities along the Boro-Thamalakane-Boteti river system, an outlet of the Okavango Delta in the Northern Botswana to establish their potential contribution to the prevalence of diarrheal diseases. | Quantitative study. | Botswana | The quality of water. How the water is stored at home. Integrated control programs focusing on improving quality of water both at source and point of use. Promotion of improved hygiene practices. | The quality of water. How the water is stored at home. |
| McGill, B.M., Altchenko, Y., Hamilton, S.K., Kenabatho, P.K., Sylvester, S.R. & Villholth, K.G. (2019). | Complex interactions between climate change, sanitation, and groundwater quality: a case study from Ramotswa, Botswana. | The study investigates the human and natural systems linking climate, sanitation, and groundwater quality in Ramotswa, a rapidly growing peri-urban area in the semi-arid Southeastern Botswana, relaying on transboundary Ramotswa aquifer for water supply. | Mixed methods | Botswana | Economic activity– Economic activity depends mainly on political willingness by the government. | Demographics Economic activity Climate change Land use |
| Mlenga, D.H. (2016). | Towards Community Resilience, Focus on a Rural Water Supply, Sanitation and Hygiene Project in Swaziland. | To assess the effectiveness of different approaches of water, sanitation, and hygiene (WASH) in reducing and mitigating against potential risk of disaster and promoting community resilience. | Mixed methods study. | Eswatini | The WASH interventions implemented by the NGOs. Improved access and availability of potable water. Improved knowledge, change attitudes and practices towards hygiene and sanitation. | Drought. Local community's resilience to the prevailing WASH challenges. Infrastructure decline. Low investment in WASH infrastructure. |
| Gwimbi, P. (2011) [28]. | The microbial quality of drinking water in Manonyane community: Maseru District (Lesotho). | To assess, at micro level the E. coli and total coliform counts in water samples from different drinking water sources in Manonyane community. A household analysis was conducted to assess the community's perception towards the quality of its water and practices aimed at protecting its sources. The study was planned to provide information that could assist in working out a model for safe drinking water supply to the community. | Cross sectional study | Lesotho | Prompt intervention to mitigate the potential health impact of water-borne diseases in the community. A proper sanitary survey and implementation of water and sanitation projects in the community. | Pollution. Poor source water protection. Poor sanitation and low level of hygiene practices. Lack of monitoring and healthcare awareness. |

*(Continued)*

**Table 1.** (Continued)

| Author(s)/year of publication | Title of the study | Objective(s) of the study | Type of the study | Country | Facilitators for WASH | Barriers for WASH |
|---|---|---|---|---|---|---|
| Gwimbi, P., George, M. & Ramphalile, M. (2019). | Bacterial contamination of drinking water sources in rural villages of Mohale Basin, Lesotho: exposures through neighbourhood sanitation and hygiene practices. | To evaluate E. coli counts in drinking water from selected communal water sources and their relationship with water source protection status and neighbourhood sanitation and hygiene practices in rural villages of Mohale Basin in Lesotho. | Cross-sectional study–mixed methods. | Lesotho | Source water protection status. Community-led sanitation and hygiene education. Improved water source protection. | Source water protection status. Contamination of water sources by e.coli. Contamination of water sources with faeces. Poor neighbourhood sanitation and hygiene condition. |
| Chunga, R.M., Ensink, J.H.J., Jenkins, M.W. & Brown, J. (2016). | Adopt or Adapt: Sanitation Technology Choices in Urbanizing Malawi. | To understand (1) why Eco sanitation uptake has been low in urban areas, and (2) how communities are meeting the challenge of increasing demands on space in sanitation technology choice. | Mixed-methods | Malawi | Pit emptying services. Construction of new pit latrines with a slab. Adaptation of locally promoted, novel sanitation technology known as ecological sanitation (ecosan). | Concerns about space for replacing pit latrines. Reluctance to unknown technology in pit latrine construction. |
| Shiras, T., Cumming, O., Brown, J., Muneme, B., Nala, R. & Dreibelbis, R. (2018). | Shared Sanitation Management and the Role of Social Capital: Findings from an Urban Sanitation Intervention in Maputo, Mozambique. | Our study sought to explore the differences in management processes between users of improved and unimproved shared latrines and investigate the determinants and impacts of collective action processes. | Qualitative | Mozambique | Developing social capital within small community units. WASH interventions employing effective collective action strategies to disseminate lessons and share behavior change tactics, e.g. electing a compound leader to implement and oversee adherence to latrine management strategies. Increased latrine ownership by individual households. Collective decision-making for shared larine users. Creating monthly financial contribution to help with ongoing latrine maintenance costs or cleaning supplies. Simple, low cost interventions informed by modern behavioral science to provide replicable approaches for increasing social capital or finding mechanisms for latrine management that rely less on complex social processes. | Shared sanitation. |

(*Continued*)

**Table 1.** (Continued)

| Author(s)/year of publication | Title of the study | Objective(s) of the study | Type of the study | Country | Facilitators for WASH | Barriers for WASH |
|---|---|---|---|---|---|---|
| Hans-Joachim, M., Mosch, S. & Harter, M. (2018). | Is Community-Led Total Sanitation connected to the rebuilding of latrines? Quantitative evidence from Mozambique. | This study investigates the effects of community-led total sanitations (CLTS) participation on latrine rebuilding and the influences of CLTS participation on personal, physical, and social context factors and psychosocial factors by conducting a cross-sectional survey in Mozambique. | A cross-sectional survey. | Mozambique | Community-Led Total Sanitation (CLTS). Latrine rebuilding depends on education, soil conditions, social cohesion, and a feeling of being safe from diarrhoea, the perception that many other community members own a latrine, and high confidence in personal ability to repair or rebuild a broken latrine. Social and psychosocial factors. | Heavy rains hit the north of Mozambique and many latrines collapsed. |
| Lewis, E.W., Nguza, S. & Selma, L. (2018). | Assessment of accessibility of safe drinking water: A case study of the Goreangab informal settlement, Windhoek, Namibia | In this study water accessibility in the Goreangab informal settlement, Windhoek, Namibia was analyzed. | Mixed methods | Namibia | Incorporation of an integrated water resource management framework and a public–private partnership to improve the settlement's water supply management. | Informal settlements. Poor water accessibility. Long distances to water sources. Water affordability. People's high reliance on contaminated water for cooking and drinking. The inability of the municipality to meet the demands of migrants flocking in search for better opportunities. |
| Abia, A.L.K., Schaefer, L., Ubomba-Jaswa, E., & Le Roux, W. (2017). | Abundance of Pathogenic Escherichia coli Virulence-Associated Genes in Well and Borehole Water Used for Domestic Purposes in a Peri-Urban Community of South Africa. | The current study was carried out to evaluate the microbial quality of wells and boreholes in Stink water, a peri-urban community of South Africa, using E. coli as an indicator organism. More importantly, the study also sought to determine the prevalence of pathogenic E. coli virulence-associated genes in these water sources so as to infer any possibility of infection from the consumption of untreated water from these water sources. | | South Africa | | Pathogenic E. coli strains. |

*(Continued)*

**Table 1.** (Continued)

| Author(s)/year of publication | Title of the study | Objective(s) of the study | Type of the study | Country | Facilitators for WASH | Barriers for WASH |
|---|---|---|---|---|---|---|
| Sibiya, J.E. & Gumbo, J.R. (2013). | Knowledge, Attitude and Practices (KAP) Survey on Water, Sanitation and Hygiene in Selected Schools in Vhembe District, Limpopo, South Africa. | The specific objectives of the study were: to understand the knowledge, attitudes and practices of learners towards water, sanitation and hygiene; to assess the availability and reliability of water supply that is used by learners at the selected secondary schools; and to assess the current status of sanitation and hand washing facilities at the selected secondary schools. | Mixed methods | South Africa | The high level of knowledge about waterborne diseases. Positive attitude and improved practices on hygiene. Urban settings. Proper handwashing facilities. Clear borehole water quality though the microbial quality was unknown. Adequate water sources. | Inadequate knowledge on transmission routes of waterborne diseases. Lack of knowledge in relation to water-based diseases and their prevention. Lack of soap at handwashing facilities. Inadequate water supply and sanitation facilities, e.g. in rural settings/schools. No handwashing areas and no sanitary bins for girls. Some schools had toilets with broken toilet doors offering no privacy. Inadequate water sources. |
| Nefale, A.D., Kamika, I., Obi, C.I. & Momba, M.N.B. (2017). | The Limpopo Non-Metropolitan Drinking Water Supplier Response to a Diagnostic Tool for Technical Compliance. | This study focused on applying the diagnostic tool for technical compliance as well as assessing the compliance of water treatment plants with management norms. | Quantitative study | South Africa | | Compliance of small water treatment plants with accepted drinking water quality standards and management norms is still a challenge in the rural areas of South Africa. Poor condition of laboratory equipment and operations. Shortage of staff, especially skilled personnel. Lack of measuring instruments/laboratory equipment, chemicals. Insufficient funds. |
| Tidwell, J.B., Chipungu, J., Chilengi, R., Curtis, V. & Aunger, R. (2019). | Theory-driven formative research on on-site, shared sanitation quality improvement among landlords and tenants in peri-urban Lusaka, Zambia | This paper reports the results of a formative research study that was designed to examine how toilets can be improved in a PUA of Lusaka, Zambia. The main objectives were to understand the existing state of sanitation, the process by which sanitation quality is maintained and improved, the roles of landlords and tenants in those processes, and the main drivers of quality maintenance and improvement. | Qualitative study | Zambia | Shared, on-site sanitation maintenance and improvement behaviors. Consumer-driven, sustainable improvements investments in toilet improvements. Introducing better shared cleaning systems. | Poor coordination among tenants–shared sanitation facilities. Lack of communication between users of shared sanitation facilities, e.g. landlords and tenants. |

(*Continued*)

**Table 1.** (Continued)

| Author(s)/year of publication | Title of the study | Objective(s) of the study | Type of the study | Country | Facilitators for WASH | Barriers for WASH |
|---|---|---|---|---|---|---|
| Psutka, R., Peletz, R., Michelo, S., Kelly, P. & Clasen, T. (2011). | Assessing the Microbiological Performance and Potential Cost of Boiling Drinking Water in Urban Zambia. | This is one of a series of studies designed to assess the microbiological effectiveness and cost of boiling as a means of treating water in the home. | Quantitative study | Zambia | Safe-storage practices to minimize recontamination. | Over-reporting and inconsistent compliance to 'cleaning' water for drinking. Lack of residual protection and unsafe storage and handling. Cost of boiling—The potential cost of fuel or electricity for boiling. |
| Thys, S., Mwape, K. E., Lefèvre, P., Dorny, P., Marcotty, T., Phiri, A.M., Phiri, I.K. & Gabriël, S. (2015). | Why Latrines Are Not Used: Communities' Perceptions and Practices Regarding Latrines in a Taenia solium Endemic Rural Area in Eastern Zambia. | The objective of this research was therefore to assess the communities' perceptions, practices and knowledge regarding latrines in a T. solium endemic rural area in Eastern Zambia, in order to identify possible barriers to their construction and use and to propose, eventually, adaptations of strategies to overcome cysticercosis, and other sanitation related diseases locally. | Qualitative– Focus group discussions | Zambia | A "people-centered" preventive approach that addresses both the perception of the disease and its management. Control strategies directed to the patterns of people's behavior associated with the phases of transmission of the disease. People's perceptions, knowledge and reported behaviors regarding the use and the construction of latrines. Seeking privacy and taboos were both identified as the key factors influencing the possession and use of sanitation facilities. Latrine promotion messages that are not only focused on health benefits. Anthropological studies for an in-depth understanding of sanitation practices within particular contexts in order to enhance the design of adapted interventions. | The existing challenges of cysticercosis control in endemic regions. People's perceptions, knowledge and reported behaviors regarding the use and the construction of latrines. |
| Tidwell, J.B., Chipungu, J., Bosomprah, S., Aunger, R., Curtis, V. & Chilengi, R. (2019). | Effect of a behaviour change intervention on the quality of peri-urban sanitation in Lusaka, Zambia: a randomised controlled trial. | To investigate to what extent sanitation could be improved by the residents of an informal settlement in Zambia themselves, through behaviour change promotion alone, in the absence of institutional change or financial subsidy. | Mixed methods | Zambia | The poor quality of toilet provision. Willingness to pay for quality improvements of toilets. | Toilets shared by multiple households. |

(*Continued*)

**Table 1.** (Continued)

| Author(s)/year of publication | Title of the study | Objective(s) of the study | Type of the study | Country | Facilitators for WASH | Barriers for WASH |
|---|---|---|---|---|---|---|
| Yeboah-Antwi, K., MacLeod, W.B., Biemba, G., Sijenyi, P., Hohne, A., Verstraete, L., McCallum, C.M. & Hamer, D.H. (2019). | Improving Sanitation and Hygiene through Community-Led Total Sanitation: The Zambian Experience. | The article presents the effect of implementing Community-Led Total Sanitation (CLTS) on sanitation and hygiene indicators in populations targeted to benefit from this package of interventions. | A pre- and post-intervention design. | Zambia | Community-led total sanitation implementation. Access to improved sanitation facilities. Reduced open defecation. Improved handwashing practices. Enhanced investment in sanitation and hygiene promotion. | |
| Ncube, F., Kanda, A., Chahwanda, M., Margaret Macherera, M. & Ngwenya, B. (2020). | Predictors of hand hygiene behaviours among primary and secondary school children in a rural district setting in Zimbabwe: a cross-sectional epidemiologic study. | The objectives of the present study were to (a) identify positive and negative hand hygiene practices, (b) ascertain the determinants for the use of desirable hand hygiene practices and (c) suggest interventions for promoting hand hygiene among school children. | A descriptive cross-sectional epidemiologic study | Zimbabwe | Investment in hand hygiene behaviour change processes. WASH promotion campaigns among school children. Empowerment of WASH clubs in schools. | |

especially in rural areas [27, 31]. Water quality and supply from many countries was reported to be compromised due to a lack of WaSH infrastructure. Some studies reported poor and inadequate protection of water sources, poor access to clean water and dependency on contaminated water from unprotected sources [30]. There were reports of water sources contamination by human excreta because of a shortage of latrines, or lack thereof. Inadequate investment in WaSH infrastructure was reflected by poor maintenance of the existing infrastructure. Geographical inequalities were identified as an existing barrier to improved drinking water supply, sanitation and hygiene particularly in rural areas of Southern Africa.

**Population growth.** It was evident that there was strain on WaSH services predominantly in urban areas where demands for WaSH services increased due to rapid population growth [25, 30]. For example, the challenge with population growth in some countries as evidenced by the inability to efficiently provide clean water services for the growing informal settlement population. In some instances, rapid population growth led to congestion thereby compromising sanitation and hygiene practices especially in places where sanitation facilities were shared. Overcrowded spaces in some countries were reported in different studies as a major factor contributing to pollution and poor neighbourhood sanitation and hygiene practices. From the studies reviewed, concerns about space/land emerged especially with regards to replacing pit latrines that filled up quickly owing to population growth.

**Inadequate knowledge on healthy WaSH practices.** People's perceptions, knowledge and reported behaviors regarding WaSH facilities such as latrines reflect their knowledge of healthy WaSH practices. Due to inadequate knowledge on the importance of improved sanitation and hygiene, some people are reluctant to change their behavior and learn how to use the introduced latrine facilities [29–31]. This was seen in places where community members practiced open defecation. Some community members were reluctant to accept and use latrines. Inadequate knowledge on the transmission of diseases associated with poor WaSH practices was reported as one of the challenges to healthy lifestyle change.

**Ineffective local community engagement.**   Effective local community engagement in interventions for WaSH practices is critical. From the studies reviewed, there is evidence that ineffective local community engagement in interventions results in a lack of monitoring and healthcare awareness [26, 27]. Engaging local community members from the design of interventions to their implementation is crucial. Some studies reviewed alluded to successful community-led total sanitation implementation resulting from effective local community engagement.

**Climate change.**   Climate change exacerbates public health issues associated with poor sanitation and hygiene practices. The findings from some of the reviewed studies reported drought as one of the influencers to barriers to improved WaSH practices. Inadequate water supply, especially during the dry seasons was described as a constraint to improved hygiene including handwashing [33]. Different countries in Southern Africa experience droughts due to climate change and that compromises WaSH practices. Among other challenges, drought seasons experienced in Southern Africa contribute to the existing challenge of disease control in endemic regions where improved WaSH facilities are most needed [25, 26]. The following themes emerged as key facilitators to WaSH practices in the region, (a) effective local community engagement, (b) increased investment on WaSH infrastructure, (c) increased latrine/toilet ownership by individual households and (d) development of social capital within small community units.

**Local community engagement.**   The reviewed studies indicated the importance of the local community's engagement in WaSH related interventions that promote improved sanitation and hygiene practices in society [16, 26, 29]. Initiatives such as community-led sanitation and hygiene were easily introduced in places where the local community members were effectively engaged [17, 27]. In places where communities used community latrines, community-led sanitation programs led to easy decision-making processes because local communities were practically engaged in interventions [21].

**Investment in WaSH infrastructure.**   WaSH infrastructure is critical for improved WaSH services. Some of the studies reviewed, from South Africa reported the benefits gained from increased investment in WaSH infrastructure [31]. Such benefits include improved access to sanitation and hygiene facilities. Investments on WaSH infrastructure also improved safe-water-storage minimizing contamination [30].

**Toilet ownership.**   The studies reviewed showed that latrine ownership by individual households played an important role in practicing healthy WaSH behaviors. Increases in individual households' ownership of a latrine reduces open defecation practice, and the use of shared latrines and promotes a healthy lifestyle [21]. The reviewed studies indicated informal settlements as some of the places at which community members struggle to maintain improved sanitation and hygiene [21, 22].

**Social capital development.**   The importance for any society to have established networks of relationships was evident in the reviewed articles. Such social capital networks contribute positively towards improved WaSH facilities and positive attitudes and behaviors [21]. The studies reviewed indicated that the development of social capital was easily established in small communities leading to effective communication essential to creating healthy living awareness in these settings.

## Discussion

Our review of published articles on WaSH practices in Southern Africa identified and analyzed facilitators and barriers to the effective implementation of WaSH. The following barrier themes emerged from the analysis: (1) geographical inequalities, (2) climate change, (3) low

investment in WaSH infrastructure, (4) low knowledge levels on waterborne diseases, (5) ineffective local community engagement. Facilitators for WaSH practices that emerged from the analysis included: (a) effective local community's engagement in WaSH interventions, (b) increased investment on WaSH infrastructure, (c) local community's engagement in WaSH interventions, (d) increased latrine ownership and (e) development of social capital within small community units.

## Geographical inequalities

While notable advances have been made in the provision of drinking water supply and sanitation worldwide [34], poor sanitation and inadequate clean drinking water supply especially in rural areas remain an important challenge in most African countries [22]. The existing barriers to improved drinking water supply and sanitation are the geographical inequalities experienced in most rural areas in Southern Africa where there are generally poor basic services provision resulting in unhealthy living conditions [29].

## Climate change

Climate change was noted as a significant challenge to water and sanitation services posing risks like damage to infrastructure due, for example, to flooding, depletion of water sources due to declining rainfall and increasing demand; and compromised water quality [35]. We noted that climate change has affected both surface and groundwater flow. Understanding the interaction between climate change, land usage, the demographic and economic activities in the region is essential in ensuring that there is water security in Southern Africa [25].

## Low investment in WaSH infrastructure

The results of the review showed that Southern Africa is among the regions with the lowest basic sanitation coverage of homes that have access to clean and safe drinking water. Poverty [19], and sharing of sanitation facilities were noted as contributing factors to poor WaSH practices in Southern Africa [21]. Insufficient investment on sanitation and hygiene resources [32] in Southern Africa contributes tremendously as a hindrance to improved WaSH practices. Addressing this requires a political will of governments to increase investments targeted to improve WaSH infrastructure. The current low investment in WaSH resources in most of the Southern African countries has led to poor implementation of water safety plans [19, 26]. Due to low investment in WaSH infrastructure, compliance of small water treatment plants to accepted standards of drinking water quality and management has resulted in inadequate provision of water supply and sanitation facilities especially in rural areas remains a challenge [19]. Rapid urbanization has added to the strain on investments that could be used to improve sanitation infrastructure in Southern Africa. We have noted that urbanization has concentrated people in areas but not matched that with sanitation development This has led to failure to meet the growing urban population's improved WaSH needs [25].

## Low knowledge levels on water borne diseases

An increase in knowledge related to water-borne diseases may contribute to a decrease in the prevalence of water-borne diseases. However, low levels of knowledge on water-borne diseases and their transmission routes have been reported in Southern Africa [31]. This may be improved through health education on the role of WaSH practices in reducing water-borne diseases [26, 36].

### Effective local community's engagement in WaSH interventions

This review indicated that effective community engagement plays a critical role in ensuring that interventions succeed [37]. Implementation challenges comprising cultural practices, possible negative attitudes and poor communication during the intervention can be eliminated through effective local community engagement. In addition to overcoming several implementation challenges, effective community engagement encourages positive attitudes in community-led intervention programs [17, 27, 32].

The major facilitators to WaSH practices in this review were: (1) increased investment on WaSH infrastructure, (2) effective local community engagement, (3) increased latrine/toilet ownership by individual households, and (4) development of social capital within small community units.

### Increased investment in WaSH infrastructure

Increased investment in WaSH infrastructure was identified as an important facilitator to improved WaSH practices [26]. Although the SDGs for safe drinking water have been achieved globally [18], many people, in rural Africa are still dependent on unsafe water sources such as rivers and unprotected wells for domestic use. Through increased investments in WaSH infrastructure, some countries in Southern Africa have improved access and availability of clean water [26] and stepped up effective promotion of hygiene practices [16], improved knowledge, attitudes and practices towards hygiene and sanitation [26]. Another benefit of increased investment for WaSH infrastructure is the improvement of water source protection [27] which is a major challenge in most Southern African communities. Furthermore, improved infrastructure can contribute toward better water storage at home [20].

### Local community's engagement in WaSH interventions

Our study findings indicated effective local community engagement in WaSH interventions as one of the important facilitators to WaSH practices [32]. Effective engagement of local communities in interventions stimulates interest in interventions and results in increased levels of knowledge on water-borne diseases [26]. Through effective engagement, community-led sanitation and hygiene education programs are easily introduced and executed [17]. Furthermore, engaging the local community assists in mobilizing the adaptation of new sanitation technologies such as ecological sanitation (ecosan) [29], a technique that makes it possible to safely use human excreta in agriculture [29]. In cases where the community uses shared latrines, effective community engagement makes promotes collective decision-making among shared larine users easier [21].

### Increased latrine ownership

Open defecation is mainly a rural phenomenon ascribed to poor latrine ownership at the community and household levels [38]. The results from the review showed that increased latrine ownership by individual households contributes to improved WaSH practices in a community [21]. Lack of sanitation facilities leads to uncontrolled disposal of household and human waste into surrounding water bodies leading to pollution and an increased risk for water-borne infections in society [18].

### Development of social capital within small community units

Developing social capital was identified as an effective strategy for health improvements especially in small communities. The development of networks of relationships among people who

lived and worked in some societies in Southern Africa enabled such communities to function effectively in facilitating improved WaSH practices [21].

## Limitations

We reviewed articles from almost all the countries in Southern Africa but limited the search of articles to only those published in English thus possibly missing experiences from some countries in the region. We may also have missed some critical literature because we only focused on literature published in peer-reviewed journals. We acknowledge that the application of filters during database search may have excluded other studies that could have been relevant to the review. Despite these limitations, we believe that our search strategy was comprehensive, and that we reviewed relevant literature in public health and the subject matter we explored.

## Conclusion

Water and sanitation are critical to ensuring healthy lifestyle. However, many people and communities in Southern Africa still lack access to safe water and improved sanitation facilities. Rural areas are the most affected by barriers to improved WaSH facilities compared to urban settings. Studies focusing on the mitigation of the existing inequalities related to WaSH developments should be conducted. Our review has shown that, the current WaSH conditions in Southern Africa do not equate to the improved WaSH standards described in the SDGs 6 on ensuring access to water and sanitation for all. Key barriers to improved WaSH practices identified include rurality, climate change, low investments to WaSH infrastructure, inadequate knowledge of water-borne illnesses and lack of community engagement. The review also identified facilitators to WaSH practices comprising social capital development, increased latrine ownership, effective local community engagement and increased investment to WaSH infrastructure. A knowledge gap exists in the continued monitoring of progress in facilitators and barriers to improved WaSH practices in the region. There is also a gap in the literature on solutions to mitigating existing barriers to improved WaSH practices in Southern Africa.

## Supporting information

**S1 Checklist. Preferred Reporting Items for Systematic reviews and Meta-Analyses extension for Scoping Reviews (PRISMA-ScR) checklist.**
(PDF)

**S1 File. Search strategy–PubMed.**
(DOCX)

**S2 File. Quality of individual studies.**
(DOCX)

**S1 Protocol.**
(DOCX)

## Acknowledgments

The authors acknowledge the input from the editors and anonymous reviewers who helped in improving the content and quality of this paper.

## Author Contributions

**Conceptualization:** Nkeka P. Tseole.

**Data curation:** Nkeka P. Tseole, Tafadzwa Mindu.

**Formal analysis:** Nkeka P. Tseole.

**Methodology:** Nkeka P. Tseole, Tafadzwa Mindu.

**Supervision:** Chester Kalinda, Moses J. Chimbari.

**Writing – original draft:** Nkeka P. Tseole.

**Writing – review & editing:** Nkeka P. Tseole, Chester Kalinda, Moses J. Chimbari.

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
