## [Decision Letter · Decision Letter 0]

21 Apr 2022

PONE-D-21-26335

Barriers and facilitators to Water, Sanitation and Hygiene (WaSH) practices in Southern Africa: a scoping review

PLOS ONE

Dear Dr. Nkeka Peter Tseole,

Thank you for submitting your manuscript to PLOS ONE. After careful consideration, we feel that it has merit but does not fully meet PLOS ONE’s publication criteria as it currently stands. Therefore, we invite you to submit a revised version of the manuscript that addresses the points raised during the review process.

Please submit your revised manuscript by  June 2, 2022. If you will need more time than this to complete your revisions, please reply to this message or contact the journal office at plosone@plos.org. Please include the following items when submitting your revised manuscript:

We look forward to receiving your revised manuscript.

Kind regards,

Balasubramani Ravindran, Ph.D

Academic Editor

PLOS ONE

Journal Requirements:

2. Please provide the full electronic search strategy for at least one database, including any limits used, such that it could be repeated

6. Please note that in order to use the direct billing option the corresponding author must be affiliated with the chosen institute. Please either amend your manuscript to change the affiliation or corresponding author, or email us at plosone@plos.org with a request to remove this option.

7. Please upload a new copy of Figure 1 as the detail is not clear. Please follow the link for more information: https://blogs.plos.org/plos/2019/06/looking-good-tips-for-creating-your-plos-figures-graphics/" https://blogs.plos.org/plos/2019/06/looking-good-tips-for-creating-your-plos-figures-graphics/

9. Please remove all personal information, ensure that the data shared are in accordance with participant consent, and re-upload a fully anonymized data set. 

Reviewers' comments:

Reviewer's Responses to Questions

**Comments to the Author**

1. Is the manuscript technically sound, and do the data support the conclusions?

Reviewer #1: Yes

Reviewer #2: Yes

2. Has the statistical analysis been performed appropriately and rigorously? 

Reviewer #1: N/A

Reviewer #2: Yes

3. Have the authors made all data underlying the findings in their manuscript fully available?

Reviewer #1: Yes

Reviewer #2: Yes

4. Is the manuscript presented in an intelligible fashion and written in standard English?

Reviewer #1: Yes

Reviewer #2: Yes

5. Review Comments to the Author

Reviewer #1: Manuscript Number: PONE-D-21-26335

Full Title: Barriers and facilitators to Water, Sanitation and Hygiene (WaSH) practices in Southern Africa: a scoping review

Short Title: Barriers and facilitators to Water, Sanitation and Hygiene practices in Southern Africa:a scoping reviews

General Comment

1. “ Rural areas are the most affected by barriers to improved WaSH facilities compared to urban settings.

Is that population growth is a barrier for rural set up or urban in southern Africa? One of the team is population growth,

Specific Comment

Abstract:

2. “ The general themes that emerged included geographical inequalities, climate change, investment on WaSH resources, low levels of knowledge on water borne diseases and ineffective local community engagement”

Is this general for barriers and facilitator for WASH, please it need to specify and indicate clearly for the reader?

Introduction:

3. The second paragraph in References citation indicated that “Inadequate water, access to improved sanitation, and hygiene (WaSH) are global health challenges affecting about one-third of the world’s population [8, 10].” since, the reference system style of the journal is Vancouver (number system) , better start with 1, then 2 , 3.4…..in an ascending manner. rather than start with 8? Like [8, 10] indicated the manuscript.

4. Paragraph 3 line four, “ However, people still lack adequate information on WaSH leading to poor sanitation and hygiene practices.” It needs citation or evidence to this sentence?

Method

5. Inclusion and Exclusion section “ We excluded reviews, i.e. systematic, scoping and meta-analysis.” It is not clear that, was there any systematic review and metanalysis in similar topic in the study setting? if that is the case, what is the importance of this review? How many did you get the three systematic, scoping, and metanalysis?

Result

6. In the eligibility criteria; “ Five (5) additional records were identified from the reference lists of eligible articles and were included for full text review” This is not clear for your inclusion, is that a grey literature ? or you already acknowledge as a limitation as included only published articles?

Discussion

7. In the first summary paragraph “The following barrier themes emerged from the analysis: (1) geographical inequalities, (2) climate change, (3) low investment on WaSH infrastructure, (4) low knowledge levels on waterborne diseases, (5) ineffective local community engagement.

In the data abstraction sheet both barriers and facilitators are put clearly, while in this paragraph Please also show the facilitators in this summary finding paragraph?

Reviewer #2: The paper is relevant and indeed brings out the key issues underlying constrains on the progressive development of the WASH sector. Below are comments: On the abstract, in these days we now refer to access not coverage, so kindly review. Regarding the geographical inequalities, kindly expand more on what you mean on that, (ie, is it access by road, or underdeveloped areas. Just checking regarding your area on in adequate knowledge on WASH practice- " in your review no paper mentioned issues of culture and norms as barriers to knowledge" I thought this could have come out. Regarding low investments, no paper mentioned the role of policies or by laws and technology that can support such interventions. " just checking if you missed, as this is a critical piece for sustainability.

If the above areas are addressed, the paper can be accepted

6. PLOS authors have the option to publish the peer review history of their article (what does this mean?). If published, this will include your full peer review and any attached files.

Reviewer #1: No

Reviewer #2: No

---

## [Author Response · Author response to Decision Letter 0]

10 Jun 2022

Your feedback was useful and it helped us to develop our manuscript. Thank you vey much.

---

## [Decision Letter · Decision Letter 1]

7 Jul 2022

Barriers and facilitators to Water, Sanitation and Hygiene (WaSH) practices in Southern Africa: a scoping review

PONE-D-21-26335R1

Dear Dr. Nkeka Peter Tseole,

We’re pleased to inform you that your manuscript has been judged scientifically suitable for publication and will be formally accepted for publication once it meets all outstanding technical requirements.

Kind regards,

Balasubramani Ravindran, Ph.D

Academic Editor

PLOS ONE

Reviewers' comments:

Reviewer's Responses to Questions

**Comments to the Author**

1. If the authors have adequately addressed your comments raised in a previous round of review and you feel that this manuscript is now acceptable for publication, you may indicate that here to bypass the “Comments to the Author” section, enter your conflict of interest statement in the “Confidential to Editor” section, and submit your "Accept" recommendation.

Reviewer #1: All comments have been addressed

Reviewer #2: All comments have been addressed

2. Is the manuscript technically sound, and do the data support the conclusions?

Reviewer #1: Yes

Reviewer #2: Yes

3. Has the statistical analysis been performed appropriately and rigorously? 

Reviewer #1: Yes

Reviewer #2: Yes

4. Have the authors made all data underlying the findings in their manuscript fully available?

Reviewer #1: Yes

Reviewer #2: Yes

5. Is the manuscript presented in an intelligible fashion and written in standard English?

Reviewer #1: Yes

Reviewer #2: Yes

6. Review Comments to the Author

Reviewer #1: Manuscript Number: PONE-D-21-26335

Full Title: Barriers and facilitators to Water, Sanitation and Hygiene (WaSH) practices in Southern Africa: a scoping review

Short Title: Barriers and facilitators to Water, Sanitation and Hygiene practices in Southern Africa: a scoping review.

The author addressed all comments in the revised version manuscript.

Reviewer #2: The manuscript is well revised and in addition the revision is well articulated in the rebuttal letter

7. PLOS authors have the option to publish the peer review history of their article (what does this mean?). If published, this will include your full peer review and any attached files.

Reviewer #1: No

Reviewer #2: No

---

## [Editor Report · Acceptance letter]

18 Jul 2022

PONE-D-21-26335R1 

Barriers and facilitators to Water, Sanitation and Hygiene (WaSH) practices in Southern Africa: a scoping review 

Dear Dr. Tseole :

I'm pleased to inform you that your manuscript has been deemed suitable for publication in PLOS ONE. Congratulations! Your manuscript is now with our production department. 

Kind regards, 

on behalf of

Dr. Balasubramani Ravindran 

Academic Editor

PLOS ONE